

# Yak rumen microbial diversity at different forage growth stages of an alpine meadow on the Qinghai-Tibet Plateau

Li Ma[1,2,3], Shixiao Xu[1], Hongjin Liu[1,2], Tianwei Xu[1], Linyong Hu[1], Na Zhao[1], Xueping Han[1,2] and Xiaoling Zhang[1,2]

[1] Northwest Institute of Plateau Biology, Chinese Academy of Sciences, Xining, Qinghai, The People's Republic of China
[2] University of Chinese Academy of Science, Beijing, The People's Republic of China
[3] Qinghai Grassland Station, Xining, The People's Republic of China

Corresponding author
Shixiao Xu, sxxu@nwipb.cas.cn

## ABSTRACT

The rumen microbiota of ruminants plays a vital role in fiber digestion, and environmental factors affect its community structure. The yak (*Bos grunniens*) is the main livestock species that inhabits the Qinghai-Tibet Plateau (QTP) at regions located at high-altitude of 3,000–5,000 m. This work investigated the rumen bacterial community of yak that grazed on the QTP during the whole year to evaluate the relationship between the rumen bacterial community and the nutrient composition of forage plant at three stages. In this study, the diversity of the rumen prokaryotic community composition was monitored in 10 full-grazing yak in an alpine meadow of the QTP. The nutrient composition of three forage growth stages was determined: re-green stage (REGY), grassy stage (GY), and withered stage (WGY). High-throughput sequencing of bacterial 16S rRNA gene was used. The results showed that the nutritive composition of the alpine meadow changed with the seasons: crude protein (CP) (13.22%) was high in forage during REGY (spring), while neutral detergent fiber (NDF) (59.00%) was high during WGY (winter). Microbial diversity and richness were highest during REGY and the average number of operational taxonomic units from 30 samples was 4,470. The microbial composition was dominated by members of Bacteroidetes (51.82%), followed by Firmicutes (34.08%), and the relative microbial abundance changed in the three forage growth stages. Unweighted UniFrac distance PcoA showed that the bacterial community structure differed between REGY, GY, and WGY. Furthermore, taxonomic groups did not present differences regarding gender in these three stages. The rumen microbiota was enriched with functional potentials that were related to ABC transporters, the two-component system, Aminoacyl-tRNA biosynthesis, and metabolism of Purine, Pyrimidine, Starch and sucrose metabolism. Significant differences were found in the composition, diversity, and function of yak ruminal microorganisms during different forage growth stages. This indicates that microbial changes in the rumen depend on changes in the forage nutritional composition. These findings provide evidence on the rumen microbial diversity of yaks in the QTP.

## INTRODUCTION

The yak (*Bos grunniens*) is the most important animal species in the QTP, China, which is the most important terrestrial ecosystem in Eurasia (*Harris, 2010*). The yak is adapted to the harsh environment of the QTP that possesses low oxygen, strong ultra-violet (UV) radiation, and limited forage resources. Natural alpine meadows in the QTP also have qualities of low temperature, high altitude, high variability in temperature and precipitation, and these factors directly affect plant productivity and nutrition (*Xue, Zhao & Zhang, 2005*; *Huang, Li & Luo, 2017*). To the growth period of grass is approximately from 100 to 150 d per year, and the dormant period lasts for about 7 months. Grasses begin to regreen each year in April, reach their peak biomass in August, and wither in November. Yaks are kept at a full-grazing style with coarse grasses as the only food in the QTP, which leads to the particular phenomenon the local herdsmen call the yak production cycle (satiate in summer, fatten in fall, thin in winter, and die in spring) (*Long et al., 1999*). To avoid the death of yaks in spring and their decrease in body weight (BW) over winter, supplementary feeding with total mixed rations of available roughage and grains is widely used as the grazing system (*Xue et al., 2017*).

It has been reported that the yak has a special molecular mechanisms to adapt to the harsh living environment in the cold and low-oxygen environment (*Chen et al., 2015a*; *Xue et al., 2017*) and yaks have adapted to the poor forage resources of the QTP. The rumen as the first chamber of the ruminant stomach, is populated by bacteria, fungi, archaea (methanogens), and protozoa with important functions in the digestion of the complex cellulolytic biomass. Ruminal microorganisms synergistically ferment plants and then provide nutrients to the host in the form of short-chain fatty acids and microbial proteins (*Huang, Li & Luo, 2017*). To enhance the fiber digestibility and feed utilization, it is necessary to obtain a deep understanding of the microorganisms in the rumen. *Firmicutes* and uncultured species were reported to have higher abundance in the yak of the QTP compared to yak at low elevation, and several bacteria (*Ruminococcus albus* and *Prevotella ruminicola*) were either not detected or had lower abundance among fibrolytic bacteria (*Ruminococcus*, *Fibrobacter*, *Clostridium*, and *Butyrivibrio*) in the yaks of the QTP (*Yang et al., 2010*; *Dan et al., 2016*; *Huang, Li & Luo, 2017*; *Xue et al., 2017*; *Xue et al., 2018*). Previous studies have shown that factors such as diet, age, species, and seasons all impact on ruminal microbes, and diet is a particularly important contributor (*Tajima et al., 2000*). It has also been suggested that unique bacterial communities are present in ruminants of the QTP (*Chen et al., 2015b*; *Huang, Li & Luo, 2017*). *Xue et al. (2017)* reported the rumen bacteria and compared the differences of microorganisms between barn fed and grazing yak in the QTP; however, these grazed yak were only studied in October, when forage biomass and nutrients peaked and did not consider the cold season. Diet and environmental conditions influence the microbial community composition; however, little is known about the bacterial community changes over time when ruminants are on similar diets, especially in the QTP where the forage differs considerably throughout a whole year due to the harshness of the environment. Some studies have been conducted on rumen microbiota in grazing yaks and sheep in QTP (*Dan et al., 2016*; *Huang, Li & Luo, 2017*; *Guo et al., 2018*). Changes in temperature and

precipitation directly affect plant yak productivity, which in turn affects animal growth and the microbiome in the body. *Bacteroidetes* and *Firmicutes* were the two predominant in the rumen of yak, these two phyla accounting for approximately 80% of the total reads, the remaining microbes involved in *Fibrobacter*, *Spirochaeta*, and *Proteobacteria* consisted of low-abundance phyla (<10% of the total reads). At the genus level, *Prevotella*, *Butyrivibrio*, *Fibrobacter*, *RC9_gut_group* and *BS11_gut_group_norank* and the unclassified bacteria were identified as the dominant genera in the rumen bacterial community (*Chen et al., 2015b*; *Peng et al., 2015*; *Huang, Li & Luo, 2017*).

Ruminal microbial composition has first been described using traditional culture-based methods (*Dehority, Tirabasso & Grifo, 1989*), which was followed by molecular studies (*Morozumi et al., 2006*; *Fernandez-Guerra et al., 2010*; *Sadet-Bourgeteau, Martin & Morgavi, 2010*; *Bekele, Koike & Kobayashi, 2011*; *Klitgaard et al., 2013*) in recent years. The rumen bacterial diversity has been substantially underestimated by traditional methods due to its anaerobiosis, which is difficult to study outside of the animal's rumen and molecular techniques based on the amplification of 16S/18S rRNA gene fragments were widely used for the study of rumen microbes. High-throughput sequencing technology has provided microbial compositions of a wide variety of different ecosystems as well as provided biological information of many microorganisms without the need for prior cultivation. Here, high-throughput sequencing of the V3–V4 region of 16S rRNA gene was used to study yak rumen bacterial community among different forage growth stages and between female and make yaks in the QTP. This study aimed to compared the composition, diversity and functions of rumen microbiota of yak under different forage growth stages and between female and make yaks. We hypothesized that documenting concurrent rumen microbiota shifts in different forage growth stage will help us establish variation trend in yak rumen community composition throughout the year. This study greatly enhanced our understanding of changes in diet corresponding shifts in rumen microbial community composition throughout the year. The knowledge of yak microbial communities in different forage growth stage can promote the understanding of rumen microbial ecosystems and improve yak productivity.

## MATERIAL AND METHODS

The experimental design and procedures were approved by the Northwest Institute of Plateau Biology, CAS-Institutional Animal Care and Use Committee (NWIPB20160302), besides this project was carried out with the permission of the local government (Qinghai province, China), so official documents and field permits are not required in this study, the China Qinghai provincial science & technology department gave these authorizations to granting us to carry out the experiment here.

### Experimental design

In the montane grassland of the Senduo Township, Guinan County, Hainan Prefecture, Qinghai Province, China, the elevation of the studied sample plot was 3,265 m. The sample plot had a latitude of 35°31′N and a longitude of 100°55′E; the annual average temperature is 1.3 °C with extended daylight times and strong solar radiation. The abundance of solar

**Table 1 Analysis of nutrient composition and dominant species of grassland in three grazing stages of alpine meadow in QTP.** The figures in the table show the nutrient composition and main dominant species of forage at different grass growth stages. All data are mean ± SD, the REGY refers to the re-green grass growth stage; GY refers to the grass growth stage; WGY refers to the withered grass growth stage.

| Nutrient content of forage grass % | Group | | | SEM | p-value |
|---|---|---|---|---|---|
| | REGY | GY | WGY | | |
| DM | 93.32 ± 0.12 | 93.00 ± 0.55 | 93.10 ± 0.48 | 0.34 | >0.05 |
| CP | 13.22 ± 1.45a | 12.20 ± 0.24a | 4.74 ± 0.87b | 0.14 | <0.01 |
| EE | 1.07 ± 0.08b | 1.50 ± 0.04a | 1.44 ± 0.05a | 0.00 | <0.01 |
| ADF | 35.53 ± 1.29a | 30.23 ± 1.37b | 33.97 ± 1.00a | 1.51 | <0.01 |
| NDF | 52.40 ± 3.50b | 57.83 ± 2.64ab | 59.00 ± 2.43a | 8.37 | 0.07 |
| Ca | 1.28 ± 0.05b | 2.18 ± 0.12a | 1.27 ± 0.08b | 0.09 | <0.03 |
| P | 0.12 ± 0.02a | 0.14 ± 0.00a | 0.04 ± 0.02b | 0.01 | <0.04 |
| The dominant species of alpine meadow in Qinghai-Tibet plateau | Mainly for those *Gramineae*, *Cyperaceae* and *Potentilla* plants | Dominant grass species are *Elymus nutans, Poa pratensi, Kobresia humili* et al., the companion grass species are *Potentill abifurca, Saussurea pulchra, Ajani atenuifolia* et al. | Dominant grass species are *Elymus nutans* | | |

energy is evidenced by the number of annual sunshine hours of 1,252–1,333 h and the forage grass growth period is 160–171 d. The cumulative ≥ 0 °C temperature is 1587.4 °C and the experimentally studied duration was 9 months (April 2017 to December 2017). Data for long-term mean climate variables as obtained from the Qinghai meteorological station. Ten healthy three-year-old QTP yaks (five males and five females) with an average weight of 108.06 ± 2.86 kg were selected, sequentially numbered, and allowed to graze. These ten yaks, without supplementary feeding during the experimental stage, were allowed access to water. Ruminal fluid was collected during the following forage growth stage: re-greening grass growth stage (REGY, spring), grass growth stage (GY, summer), and withered grass growth stage (WGY, winter), and alpine meadow samples were collected during the same three stages. The main forage species and their relative proportions in montane grassland during these three periods are listed in Table 1. Biomass was highest in July at 146.56 g/m² and the ratio of high-quality forage to the total above-ground biomass was 55–61%. The proportion of dominant species of the forage were *Elymus nutansv* (74.5%), *Kobresia humili* (6.8%), and *Poa pratensiv* (2.0%), while further species include *Potentill abifurca* (3.4%), *Saussurea pulchra* (2.7%), and *Ajania tenuifolia* (1.4%) in GY. The proportion of dominant species of the alpine meadow in REGY were *Gramineae* (40.9%), *Cyperaceae* (36%), *and Potentilla* (2.04%) plants, and the dominant grass species was *Elymus nutans* (78.56%) in WGY (Table S1).

## Plant sample collection

Forage samples were collected by quadrats (one m × one m) from grass on which the animals grazed during the regreen stage (May 2, 2017), the grass stage (July 12, 2017), and the withered grass stage (December 7, 2017). Ten quadrats (one m × one m), of which the distance between plots exceeded 10 m, were randomly placed in the alpine meadow to collect grass samples and cut out the ground part of the quadrats with scissors at three sample collection times. Samples from each quadrat were used to investigate dominant species, dried in a 60 °C oven for 24 h to constant weight at the laboratory, and were ground in a mill and passed through a 1-mm sieve for further analysis.

## Rumen fluid collection

The rumen contents (30 samples) were collected using an esophageal tube vacuum pump sampling device (Anscitech Company, Wuhan, China). This method has been used in previous studies (*Ramos-Morales et al., 2014*; *Xue et al., 2017*; *Sun et al., 2018*), and they have shown that the stomach tube and rumen cannulation methods do not have any influence on the results. The rumen contents were filtered using four layers of gauze and approximately 75 mL of fluid was collected from each animal prior to grazing in the morning. Twenty mL of fluid was aliquoted for use and 2 mL was used for DNA extraction and subsequent sequencing. After aliquoting, the fluid samples were immediately placed into liquid nitrogen (−80 °C) for storage and transported to the laboratory for further use.

## Determination of plant nutrient compositions

The concentration of acid detergent fiber (ADF), and neutral detergent fiber (NDF) in 30 samples (10 samples per period) were determined using the Van Soest method (*Van Soest, Robertson & Lewis, 1991*). Dry matter (DM), crude protein (CP), crude fat ether extract (EE), Ca, and P were measured using AOAC methods (*Cunniff, 1995*).

## DNA extraction and polymerase chain reaction (PCR) amplification

Microbial DNA was extracted from samples using the E.Z.N.A. stool DNA Kit (Omega Biotek, Norcross, GA, USA) according to the manufacturer's instructions. The 16S rDNA V3–V4 region of the Prokaryotic ribosomal RNA gene was amplified by PCR (95 °C for 2 min, followed by 27 cycles at 98 °C for 10 s, 62 °C for 30 s, 68 °C for 30 s, and a final extension at 68 °C for 10 min using primers 341F: CCTACGGGNGGCWGCAG; 806R: GGACTACHVGGGTATCTAAT (*Guo et al., 2017*)). Amplicons were extracted from 2% agarose gels and purified using the AxyPrep DNA Gel Extraction Kit (Axygen Biosciences, Union City, CA, USA) according to the manufacturer's instructions and quantified using QuantiFluor-ST (Promega, USA). According to the standard protocols, Purified amplicons were sequenced on an Illumina Hiseq2500 platform (Illumina San Diego, CA, USA), which was conducted by Genedenovo Bioinformatics Technology Co., Ltd. (Guangzhou, China) using 250 bp paired end reads.

## Data analysis

Sequences were sorted based on their unique barcode and then, barcode and primer sequence were removed using QIIME (version 1.9.0). FLSAH (version 1.2.11) was

used to merge paired end clean reads as raw tags (*Magoc & Salzberg, 2011*). Low quality reads were eliminated to obtain high-quality clean tags by QIIME (version 1.9.0) (*Caporaso et al., 2010*). Clean tags were compared to the Gold database (http://drive5.com/uchime/uchime_download.html) using the UCHIME algorithm (http://www.drive5.com/usearch/manual/uchime_algo.html) to eliminate chimera sequences and effective tags were obtained for further analysis. These effective tags were clustered into operational taxonomic units (OTUs) of ≥ 97% similarity using the UPARSE pipeline (*Edgar, 2013*). Representative sequences were classified into organisms using the RDP classifier (*Wang et al., 2007*) based on the SILVA database (*Quast et al., 2013*).

### Statistical analysis

Alpha diversity analysis including Chao1, Shannon, Simpson, ACE, and Good-coverage were calculated by QIIME (version 1.9.0). Alpha index comparison among groups was computed by Kruskal–Wallis test in R (version 3.4.3). A principal coordinate analysis (PCoA) and ANOSIM were used to confirm findings from the distance matrices. Pearson correlations were conducted using R software (*R Core Team, 2017*) with the vegan package on the online omicsmart cloud platform (http://www.omicsmart.com/). Pearson correlations using the pheatmap package were used to analyze the relationships between environmental factors and bacteria.

Relative differential abundance tests among REGY, GY, and WGY were also conducted at phylum, genus, and family levels using Kruskal–Wallis test in R. Tukey-Kramer post-hoc test was used in conjunction with analysis of Student's $t$-test in STAMP (version 2.1.3) to compare the abundance changes between female and male yaks. Pearson correlations in the omicsmart cloud platform was used to examine the relationships among the 20 known prokaryotic genera with six forage nutrient compositions, redundancy analysis in Canoco (version 5.0) was used to examine relationships of the top 10 known prokaryotic phyla with six forage nutrient compositions. Nutrient compositions of the forage were analyzed using a general linear model, and a calculated $P$-value <0.05 was considered to indicate statistical significance. The differences between diversity and richness mean values were analysed using two-way anova test conducted with SAS 9.4 software (SAS Inst. Inc., Cary, NC, USA). Tax4fun software was used to compare the species compositions obtained from the 16S sequencing data and then to infer the functional gene composition of samples. The functional composition was predicted from the Kyoto Encyclopedia of Genes and Genomes (KEGG) database (*Minoru et al., 2012*). The KOs (KEGG orthology groups) at different forage growth stages were further examined by Kruskal–Wallis test in R.

### Accession number

The sequencing data for the 16S rRNA genes are publicly available in the NCBI (https://www.ncbi.nlm.nih.gov/) Short Read Archive under accession number PRJNA504932.

## RESULTS AND ANALYSIS

### Analysis of forage nutrient composition

The nutrient composition of mixed forage samples collected during different forage growth stages from the QTP was determined (Table 1). The CP content was the highest during

REGY and was significantly ($P < 0.05$) higher than during WGY, the REGY presented 1.0% more protein than GY. In GY, the EE content was highest and was significantly ($P < 0.05$) higher than during REGY, the GY was 0.43% higher than REGY. ADF reached 35.53% in REGY that higher than other two stages. NDF gradually increased in the trend: REGY <GY <WGY, the NDF content was highest in WGY and was significantly ($P < 0.05$) higher than REGY. Ca and P were highest during GY and the Ca content was significantly higher during GY ($P < 0.05$) compared to REGY and WGY. The main dominant species during each grazing period are shown in Table 1. The nutrient composition of herbage varies in 3 forage growth stages. The nutritive value and palatability of herbage in the REGY was higher than WGY, that is, the CP content was increased with NDF decline in REGY, whereas the NDF content was increased with CP decline in WGY.

## Sequencing results and diversity analysis of forage samples

By sequencing the V3–V4 of the 16S rRNA gene of rumen bacteria collected during the three forage growth stages, 3,104,385 raw sequences were obtained from 30 samples. After filtering low-quality sequences, the total number of effective sequences was 3,059,694. All effective tags of 30 samples were clustered into 13,280 OTUs at the similarity >97%, averaged at 4,447 OTUs per sample. There were 7,064 OTUs shared among REGY, GY, and WGY (Fig. 1, Table S2); these 7,064 OTUs accounted for 53% of all 13,280 OTUs and *Bacteroidetes*, *Firmicutes*, *Verrucomicrobia*, and *Proteobacteria* accounted for 23.56%, 17.90%, 3.28%, and 2.44% of all 13,280 OTUs, respectively. The diversity and richness indices of all samples from the three groups and genders were calculated. The number of species observed in the rumen fluid was significantly different among three grass growth stage ($p$-value = 0.03), with the highest number in female samples from REGY. Index ACE observed in the rumen fluid was significantly different among three grass growth stage ($p$-value = 0.02), with the highest number in female samples from REGY (Table 2).

## Composition of rumen bacterial community

At the phylum level, the top ten phyla were accounted for nearly 99% of all sequences. The average abundance of the three groups in 30 samples were *Bacteroidetes* (51.82%), *Firmicutes* (34.08%), *Verrucomicrobia* (5.97%), *Proteobacteria* (1.92%), *Spirochaetae* (1.12%), *Lentisphaerae* (1.07%), *Tenericutes* (0.93%), *SR1* (0.92%), *Cyanobacteria* (0.62%), and *Fibrobacteres* (0.35%) (Table. S2). *Firmicutes* had the highest relative abundance during GY, while *Verrucomicrobia*, *Lentisphaerae*, *Cyanobacteria*, and *Fibrobacteres* gradually increased in the trend, following: GY < REGY < WGY. Among the top 10 phyla, Kruskal–Wallis test demonstrated the abundance of *Firmicutes* ($p$-value < 0.01), *Verrucomicrobia* ($p$-value < 0.01), *Proteobacteria* ($p$-value < 0.01), *Lentisphaerae* ($p$-value < 0.01), and *Fibrobacteres* ($P < 0.01$) changed significantly at REGY, GY, and WGY, At the phylum level, *Firmicutes* was significantly more abundant in GY than in REGY ($p$-value <0.05) or WGY ($p$-value < 0.05). *Verrucomicrobia* was significantly lower in GY than in REGY ($p$-value < 0.01) and WGY ($p$-value < 0.01), *Proteobacteria* was significantly lower in REGY than in GY ($p$-value < 0.01) or WGY ($p$-value < 0.01), *Lentisphaerae* was significantly higher in WGY than in REGY ($p$-value < 0.01) or GY ($p$-value < 0.01) (Fig. 2).

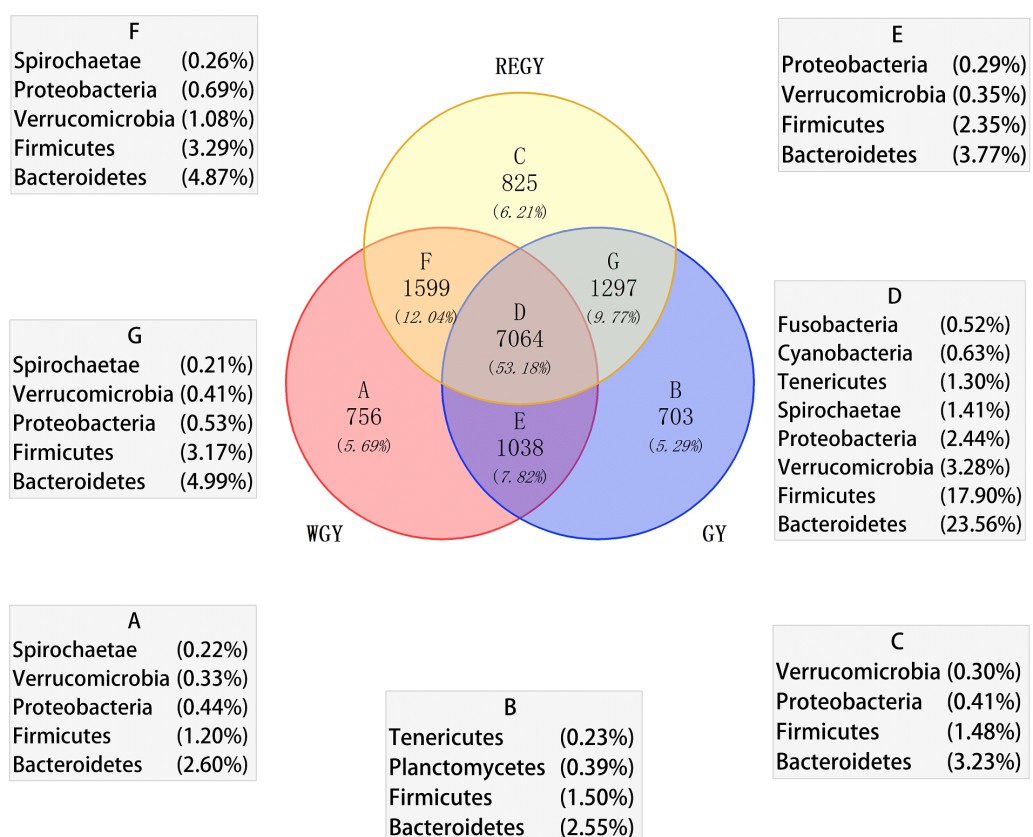

**Figure 1** **Venn data showing shared OTUs among different forage growth stage and unique to each of them.** Only OTUs at phylum are shown.

**Table 2** **Rumen microbiota diversity and richness between different grass growth stages and gender.** The data in the table indicates the effects of different grass growth stages and gender on the rumen micobiota diversity and richness index.

| Grass stage | Gender | Sobs | Simpson | Shannon | Chao | Ace |
|---|---|---|---|---|---|---|
| REGY | Female | 4657.63 | 1.00 | 10.03 | 6392.91 | 6232.83 |
|  | Male | 4609.61 | 0.99 | 9.77 | 6419.18 | 6218.95 |
| GY | Female | 4489.22 | 1.00 | 9.84 | 6201.51 | 6089.73 |
|  | Male | 4074.60 | 0.99 | 9.48 | 5712.25 | 5604.42 |
| WGY | Female | 4512.41 | 1.00 | 9.97 | 6157.27 | 5974.50 |
|  | Male | 4340.22 | 1.00 | 9.94 | 5900.43 | 5754.64 |
|  | SEM | 121.24 | 0.004 | 0.198 | 144.13 | 135.027 |
|  | $p$-value |  |  |  |  |  |
| Grass stage |  | 0.03 | 0.48 | 0.31 | 0.01 | 0.02 |
| Gender |  | 0.04 | 0.33 | 0.19 | 0.05 | 0.04 |
| Grass stage × Gender |  | 0.32 | 0.56 | 0.69 | 0.22 | 0.24 |

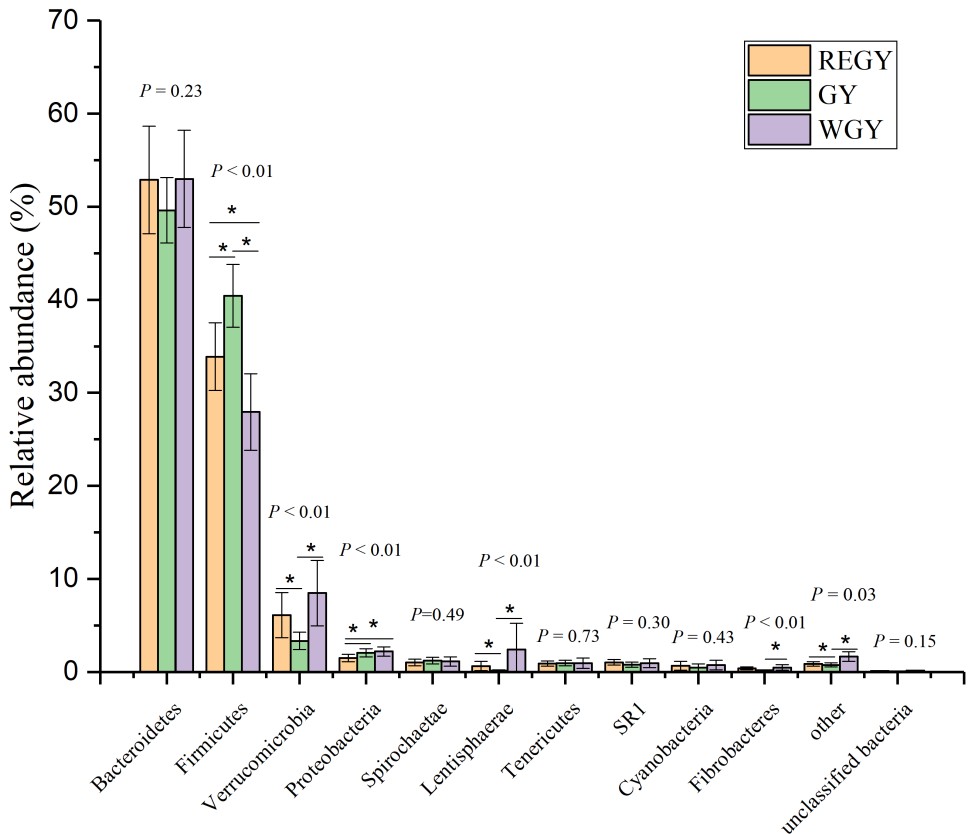

**Figure 2** **Differences of the relative abundance of dominant bacteria at phylum level in different grass growth stages (REGY, GY and WGY).** The abundance of the top 10 phyla at different grass growth stages. The significant difference in groups by Kruskal–Wallis test with $p < 0.05$ marked, the extremely significant difference in groups by Kruskal–Wallis test with $p < 0.01$ marked. Student's $t$-test was used to compare the abundance changes each two groups (REGY vs. GY, REGY vs. WGY, GY and WGY). The one-asterisk above the column represents that there is significant difference in between groups ($P$-value $< 0.05$).

At the class level, *Clostridia* (*p_Firmicutes*), *Negativicutes* (*p_Firmicutes*), and *Erysipelotrichia* (*p_Firmicutes*) all were significantly more abundant in GY than in REGY or WGY just as *Firmicutes*. *Deltaproteobacteria* (*p_Proteobacteria)* was significantly more abundant in GY than in REGY ($p$-value = 0.006) or WGY ($p$-value < 0.013) (Fig. S1A). At the order level *Victivallales* (*c_Lentisphaeria, p_Lentisphaerae*), *Oligosphaerales* (*c_Oligosphaeria, p_Lentisphaerae*), *Gastranaerophilales* (*c_Melainabacteria, p_ Cyanobacteria*) were *significantly* higher in WGY than in REGY and GY (Fig. S1B).

At the family level, the top 10 families accounted for 81.04% of the total sequences (Fig. 3, Table S3). Among these, family *Prevotellaceae* (*p_ Bacteroidetes*) (REGY: 24.76%, GY: 21.37%, WGY: 26.09%) was the most dominant family, followed by *Ruminococcaceae (p_ Firmicutes)* (REGY: 14.19%, GY: 16.50%, WGY: 13.56%), *Rikenellaceae (p_ Bacteroidetes)* (REGY: 11.59%, GY: 11.44%, WGY: 11.96%), and *Lachnospiraceae* (*p_Lentisphaerae*) (REGY: 12.62%, GY: 12.06%, WGY: 8.50%). Other families with lower abundance were *Bacteroidales_BS11_gut_group* (REGY: 9.85%, GY: 9.10%, WGY: 7.39%),
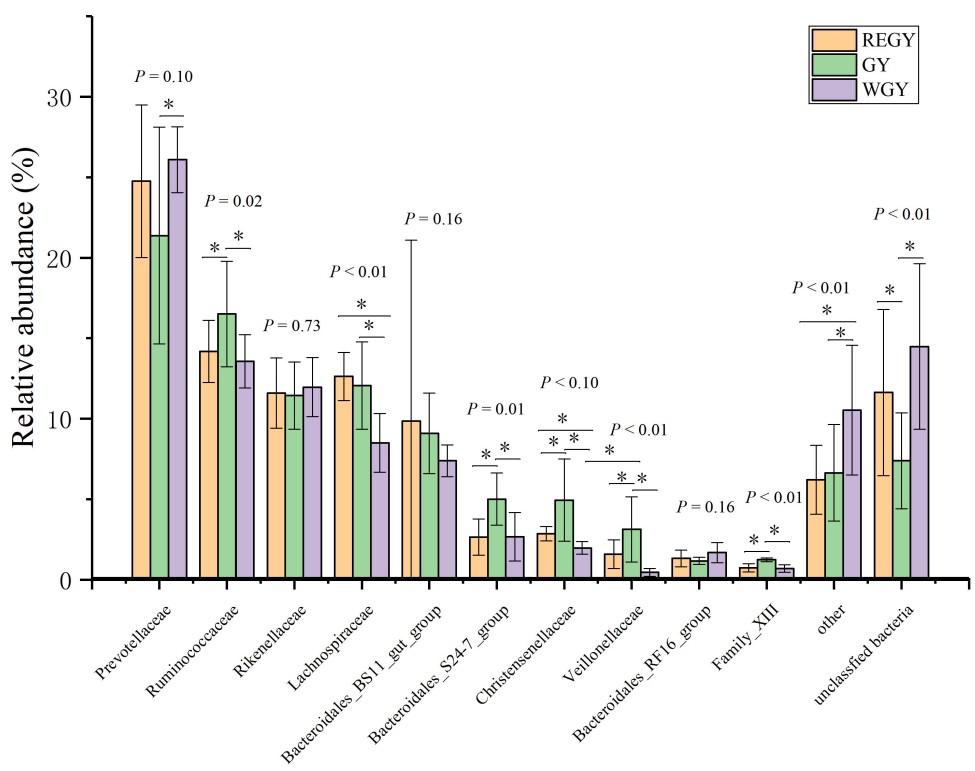

**Figure 3** **Differences of the relative abundance of dominant bacteria at family level in different grass growth stages (REGY, GY and WGY).** The abundance of the top 10 families at different grass growth stages.

*Bacteroidales_S24-7_group* (REGY: 2.64%, GY: 5.00%, WGY: 2.67%), *Christensenellaceae (p_Firmicutes)* (REGY: 2.86%, GY: 4.95%, WGY: 1.97%), *Veillonellaceae (p_Firmicutes)* (REGY: 1.58%, GY: 3.12%, WGY: 0.45%), *Bacteroidales_RF16_group* (REGY: 1.33%, GY: 1.17%, WGY: 1.69%), and *Family_XIII (p_Firmicutes)* (REGY: 0.74%, GY: 1.24%, WGY: 0.69%). Kruskal–Wallis test demonstrated that the abundances of *Ruminococcaceae* ($p$-value < 0.05), *Lachnospiraceae* ($p$-value < 0.01), *Bacteroidales_S24 -7_group* ($p$-value < 0.05), *Christensenellaceae* ($p$-value < 0.01), *Veillonellaceae* ($p$-value < 0.01), and *Family_XIII* ($p$-value < 0.01) changed significantly at all three groups (Fig. 3). At the family level, *Ruminococcaceae*, *Bacteroidales_S24-7_group*, *Christensenellaceae*, *Veillonellaceae*, and *Family_XIII* all were significantly higher in GY than in REGY. *Lachnospiraceae* was significantly lower in WGY than in REGY ($p$-value < 0.01) or GY ($p$-value < 0.01).

At the genus level, taxa with a relative abundance of ≥ 1% in at least one sample were analyzed. Sixteen taxa exhibited significantly different abundances between three groups were particularly dominant, including *Christensenellaceae_R-7_group* ($p$-value < 0.01), *Prevotellaceae_UCG-003* ($p$-value < 0.01), *Prevotellaceae_UCG-001* ($p$-value < 0.05), *Butyrivibrio_2* ($p$-value < 0.01), *Ruminococcaceae_UCG-010* ($p$-value < 0.01), *Ruminococcaceae_UCG-005* ($p$-value < 0.01), *Ruminococcaceae_UCG-014* ($p$-value < 0.01), *Ruminococcaceae_NK4A214_group* ($p$-value < 0.01), *Selenomonas_1 (f_Veillonellaceae)*,

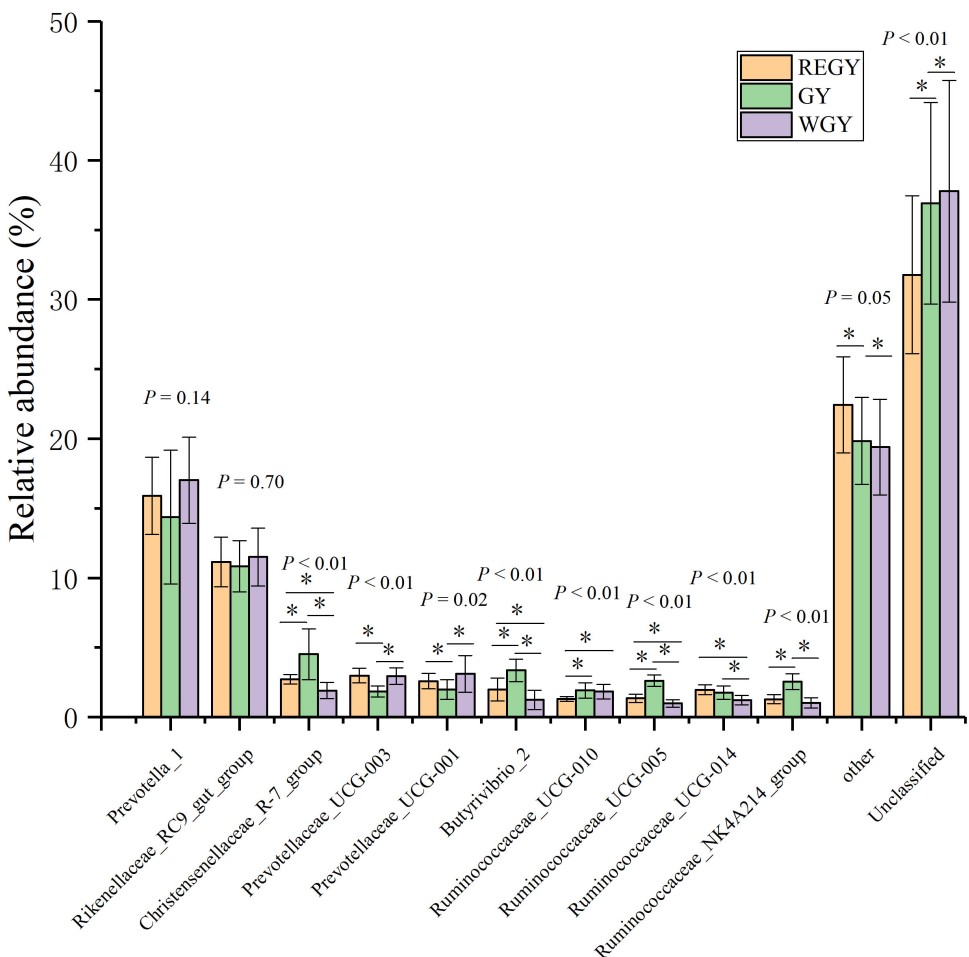

**Figure 4** **Differences of the relative abundance of dominant bacteria at genus level in different grass growth stages (REGY, GY and WGY).** The genera of the top 10 at different grass growth stages.

*Ruminococcus_1 (f_Ruminococcaceae)*, *Saccharofermentans (f_Ruminococcaceae)*, and *Pseudobutyrivibrio (f_Lachnospiraceae)*. *Prevotella_1* (15.76%), and *Rikenellaceae_RC9_gut_group* (11.17%) (Table S3). At the genus level, *Christensenellaceae_R-7_group*, *Butyrivibrio_2*, *Ruminococcaceae_UCG-005*, *Ruminococcaceae_NK4A214_group* all were significantly ($p$-value < 0.01) more abundant in GY than in REGY or WGY. *Prevotellaceae_UCG-003* and *Prevotellaceae_UCG-001* were lowest abundance in the GY compared with other two groups (Fig. 4, Table S3).

Furthermore, Student's $t$-test was used to assess significant differences in species abundance between female and male yak populations at three groups at phylum, family, and genus levels (Fig. 5). Two phyla exhibited significantly different abundances between female and male yaks, *Tenericutes* (REGY) was more abundant in the female group at ($p$-value < 0.05), and *Bacteroidetes* (WGY) was more abundant in the male group at $P < 0.05$. Two families including *Bacteroidales_S24-7_group* and *Family_XIII* were more abundant in the female yak group during GY ($p$-value < 0.05), and *Prevotellaceae*

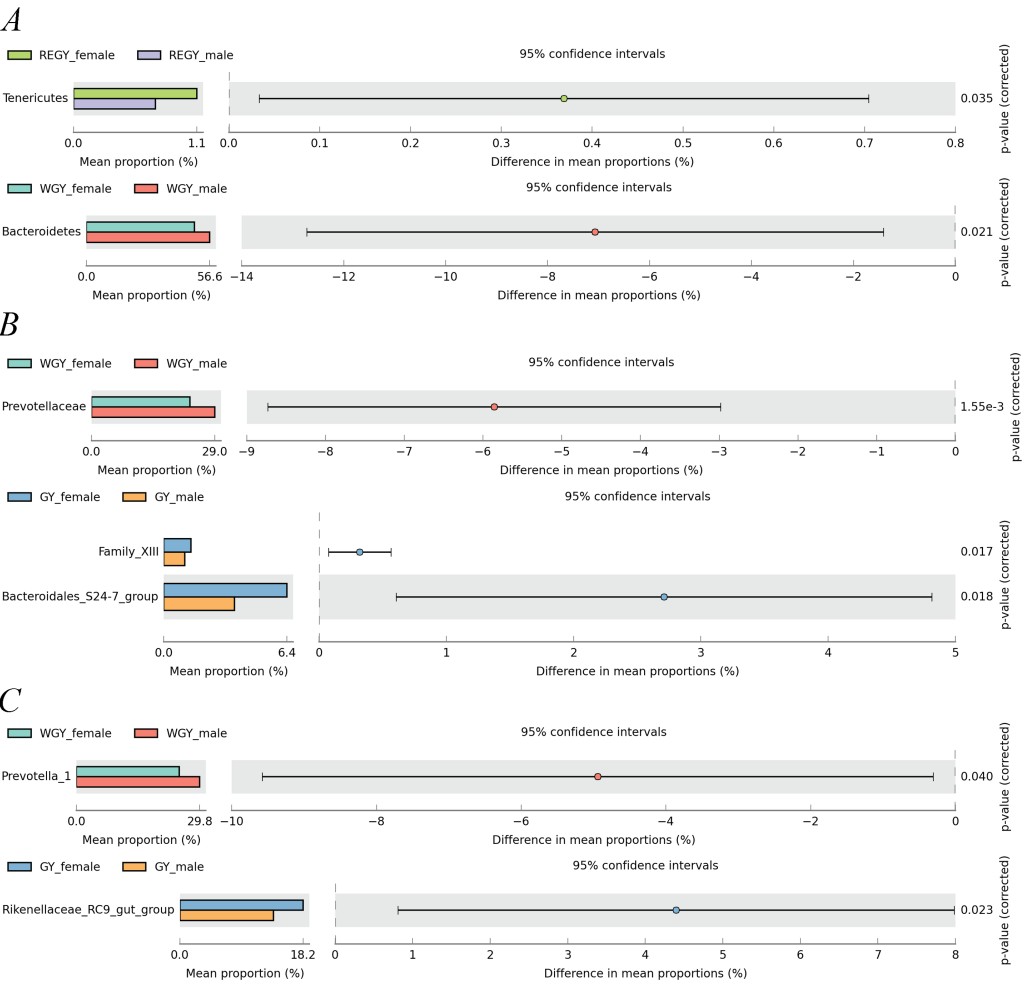

**Figure 5   Percent composition and significance of phyla (A), families (B) and genera (C) between female and male yak among different grass growth stages (REGY, GY and WGY).** Student's *t*-test was used to compare the abundance changes between female and male yaks. Only differences for which *p*-value < 0.05 are reported.

were more abundant in the male yak group during WGY (*p*-value < 0.01). Moreover, two genera including *Prevotella_1* (WGY_female: 15.07%, WGY_male: 18.99%) and *Rikenllaceae_RC9_gut_group* (GY_female: 12.02%, GY_male: 9.68%) exhibited significantly different abundances between female and male yak groups (*p*-value < 0.05).

## Microbial community analysis at three forage growth stages

PCoA was used to compare the bacterial community structure among different forage growth stages (Fig. 6). Unweighted UniFrac distance PcoA showed that the GY group community structure differed significantly from that of the other two groups (ANOSIM: Unweighted unifrac, *p*-value = 0.001, *R*-value = 0.662). Significant differences (based on ANOSIM tests) were also found between REGY and WGY groups (*p*-value = 0.001, *R*-value = 0.282). It was not detected differences in PcoA between female and male samples.

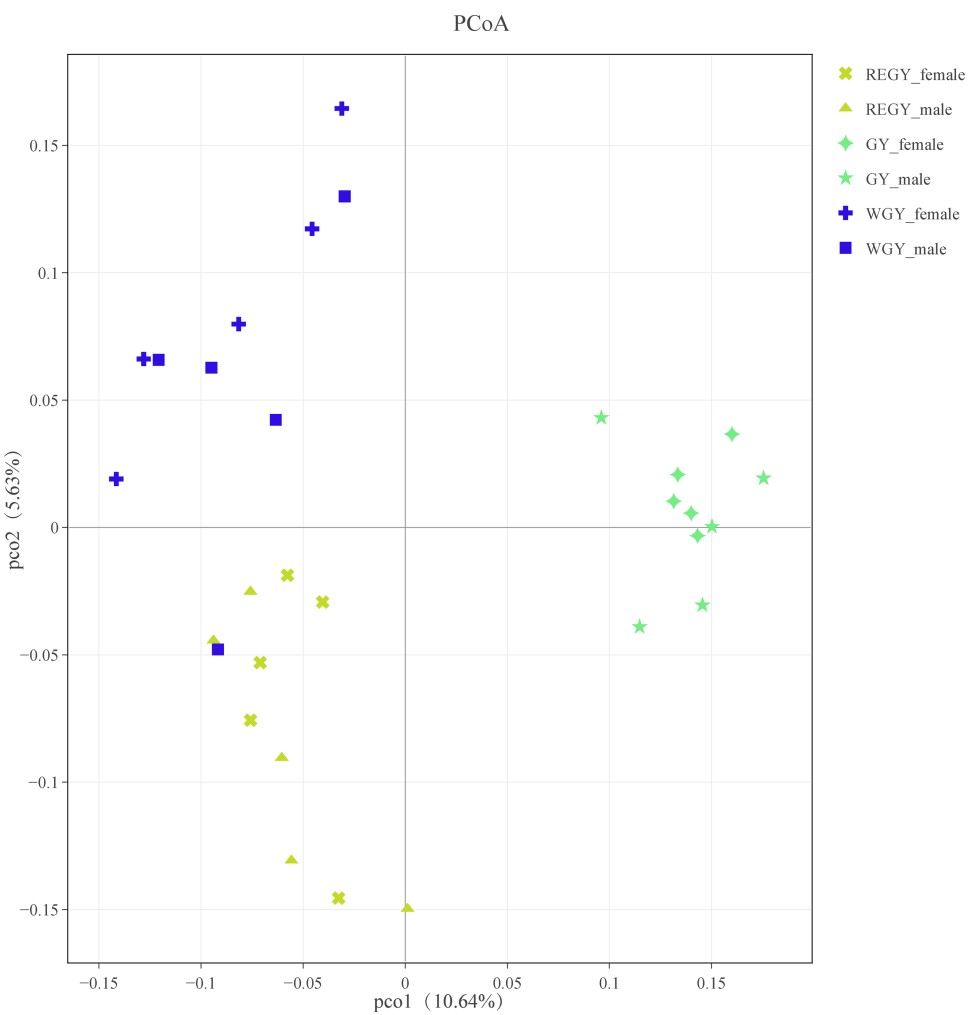

**Figure 6 Principal coordinate analysis (PCoA) of bacterial community structures of the ruminal microbiota among different grass growth stages.** PCoA plots were constructed using the unweighted UniFrac method, the non-parametric permutational MANOVA-based statistical tests adonis and ANOSIM. The REGY (yellow cross represents female yak, yellow triangles represents male yak), GY (green diamond represents female yak, green star represents male yak) and WGY (blue cross represents female yak, blue squre represents male yak) group.

## Correlations of microbial communities with forage nutrient composition

To study the relationships of the bacterial community structure with forage nutrient parameters, correlations were investigated using both Pearson correlations and redundancy analysis. Figure 7A shows that CP correlated positively with relative abundance of *Christensenellaceae_R-7_group*, *Butyrivibrio_2*, *Ruminococcaceae_UCG-005*, *Ruminococcaceae_UCG-014*, *Ruminococcaceae_NK4A214_group*, *Selenomonas_1*, and *Pseudobutyrivibrio* ($r$-value > 0.4, $p$-value < 0.05), but correlated negatively with the *Coprostanoligenes_group* ($r$-value < −0.4, $p$-value < 0.05). NDF correlated negatively with the relative abundances of *Ruminococus_1*, *Saccharofermentans*, *Prevotellaceae_UCG-003*,

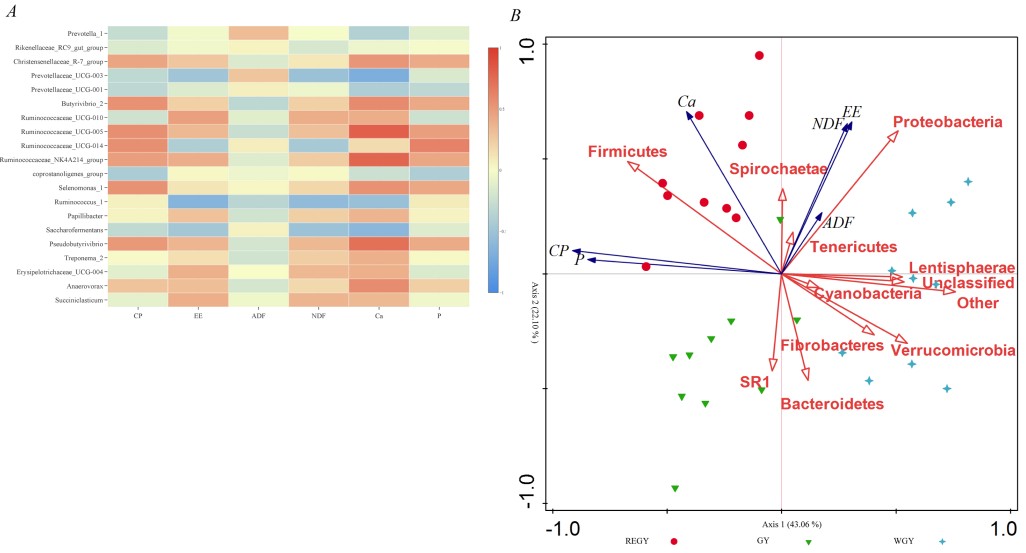

**Figure 7** **Correlations analyses between the relative abundances of bacteria genera and forage nutrient composition parameters.** (A) Pearson correlations analyses using the pheatmap package, the result numerical matrix is visually displayed in a heatmap diagram. The color depth indicates the size of the data value (correlation value). Red indicates that the correlation coefficients ($r$) were $>0.4$ and the $P$-values were $< 0.05$, blue indicates that the correlation coefficients ($r$) were $< -0.4$ and the $P$-values were $< 0.05$. (B) Redundancy analysis of the top 10 known prokaryotic phyla in association with six forage nutrient composition parameters.

and *Ruminococcaceae_UCG-014* ($r$-value $<-0.4$, $p$-value $< 0.05$). Ca correlated positively with the relative abundances of *Ruminococcaceae_UCG-005* and *Pseudobutyrivibrio* ($r$-value $> 0.4$, $p$-value $< 0.05$), but negatively with relative abundances of *Ruminococcaceae_UCG-003*, *Ruminococcus_1*, and *Saccharofermentans* ($r$-value $<-0.4$, $p$-value $< 0.05$) (Table S4).

Redundancy analysis indicated the correlation between the bacterial community structure at the phylum level and forage nutrition parameters, the two axes explained 65.07% of the differentiation of the microbial community. CP, P, Ca, NDF, and EE all correlated positively with *Firmicutes*, however correlated negatively with *Bacteroidetes*. *Proteobacteria* correlated positively with NDF and EE, *Spirochaetae* correlated positively with NDF, Ca, and EE (Fig. 7B), *Verrucomicrobia*, *Lentisphaerae*, and *Fibrobacteres* all correlated negatively with CP.

## Tax4fun gene function prediction

Yak rumen liquid bacterial community functional predictions in the three forage growth stages were investigated using Tax4fun4 and differences in KEGG and KO abundances between groups were identified. KOs in level 2 (Table S5) suggest that the pathways related to membrane transport, translation, and carbohydrate metabolism were enriched in REGY, GY, and WGY groups. At the pathway level, compared to GY group (Level 2 KOs, student's-t test, $p < 0.05$, Fig. 8), genes involved in cell growth and death, cell motility, membrane transport, and carbohydrate metabolism were significantly more abundant in the REGY group, while genes that included folding, sorting, and degradation, cell growth

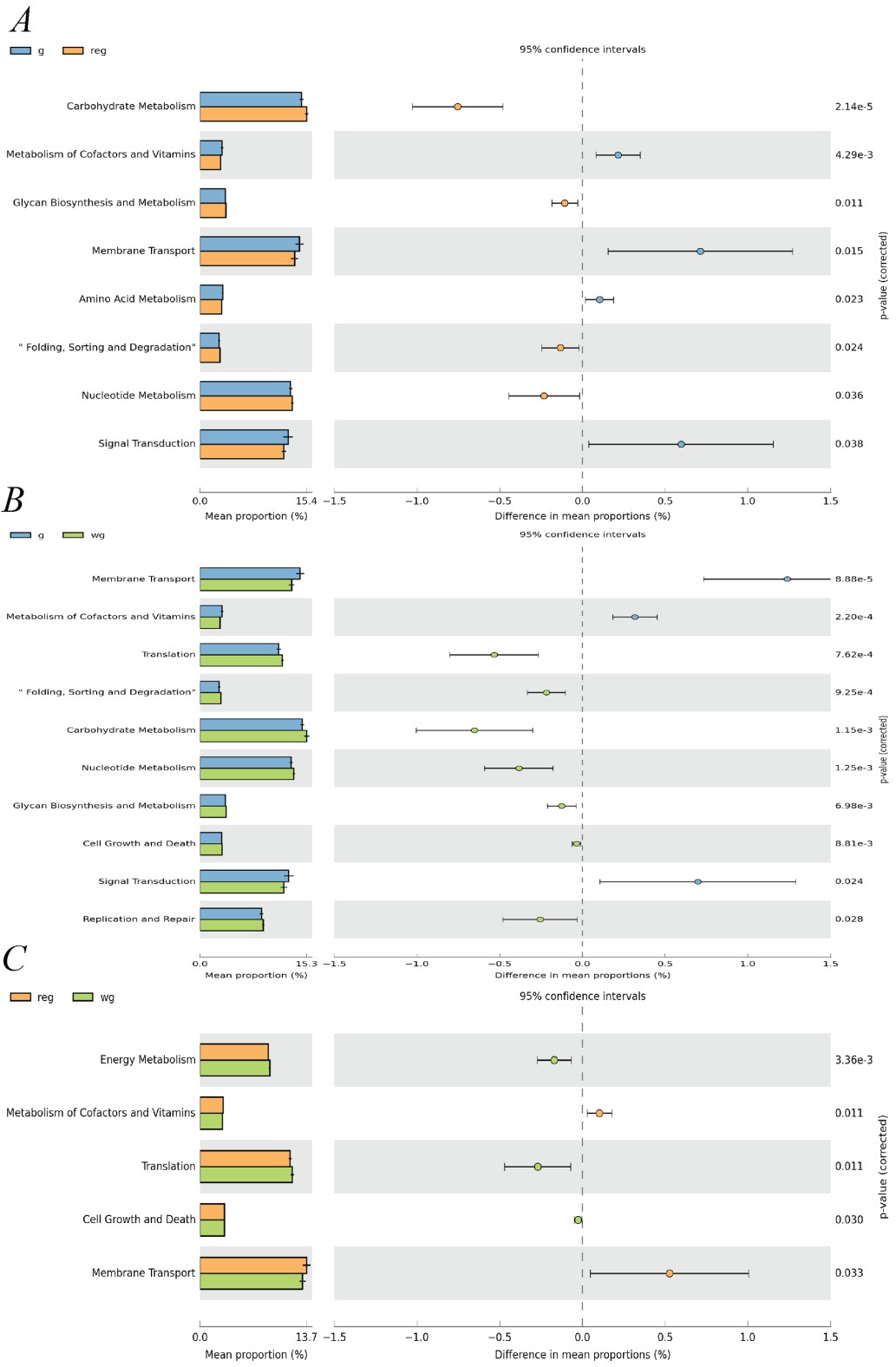

**Figure 8** **Relative abundance of predicted functions and significance of KOs in KEGG level 2 among three grass growth stage (REGY, GY and WGY).** A (REGY vs. GY); B (REGY vs. WGY); C (GY and WGY). Student's *t*-test was used to compare the abundance changes each two groups; only differences for which *p*-value < 0.05 are reported.

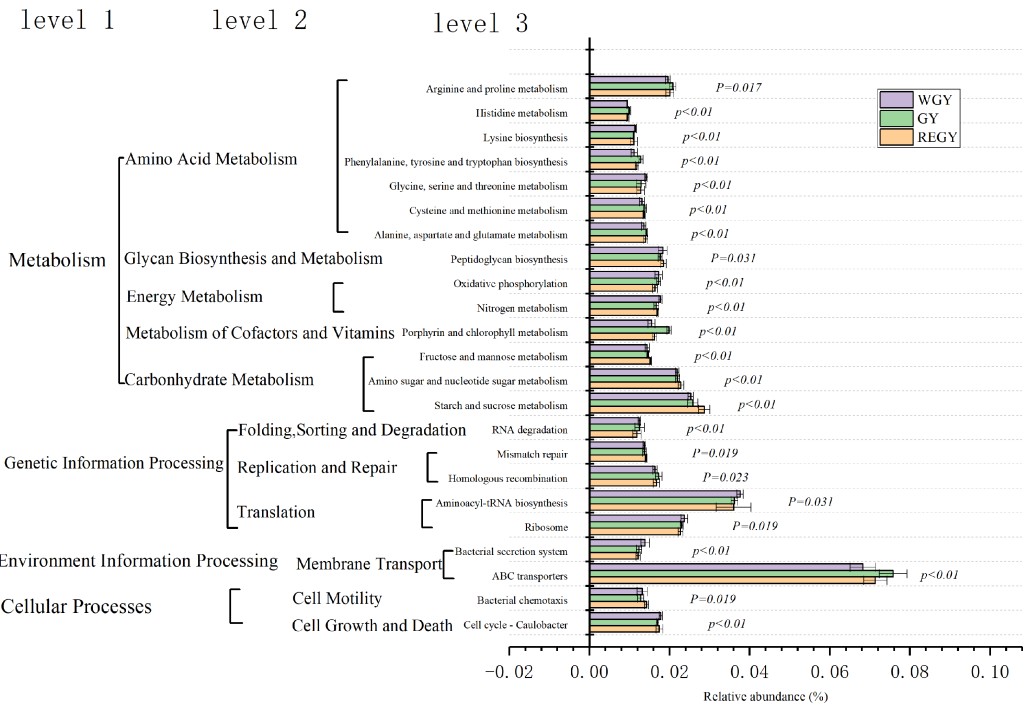

**Figure 9** **Functional predictions for rumen microbiota that significantly different KEGG (Kyoto Encyclopedia of Genes and Genomes) pathways ($p < 0.05$) of three grass growth stage (REGY, GY and WGY).** KEGG pathway at Level 1, Level 2 and Level 3 were represented. REGY, regreen grass stage; GY, grassy grass stage; WGY, withered grass stage.

and death, Translation, Glycan biosynthesis, and Metabolism and energy metabolism were significantly lower than the WGY ($p$-value $< 0.05$). The relative abundance of Metabolism of cofactors and vitamins was significantly higher in GY than in WGYS groups ($p$-value $< 0.05$).

The relative abundance of ABC transporters (7.138%) was highest in the three stages, Aminoacyl-tRNA biosynthesis (3.66%), Starch and sucrose metabolism (2.66%), Ribosome (2.32%) as well as Amino Sugar and nucleotide sugar metabolism (2.22%) were second-most abundant KOs. Kruskal–Wallis test demonstrated that a total of 181 KOs showed significant difference at REGY, GY, and WGY groups ($p$-value $< 0.05$) in the level three KEGG pathways, 23 KOs (relative abundances >1%) of which are shown in Fig. 9. The heatmap (Fig. S2) also indicated that the relative abundance of Fructose and mannose metabolism, Alanine, aspartate and Glutamate metabolism, and Cysteine and methionine metabolism in WGY group were higher than in the other two groups.

## DISCUSSION

According to prior analyses (*Chen et al., 2008*; *Latham et al., 2018*), low-quality diets in low protein supplementation had greater bacterial diversity, which could be interpreted as a further feature of the diet that contains more secondary plant compounds in low-quality diets compared to high-quality diets. (*Belanche et al., 2012*; *Fernandes et al., 2014*) also

reported that a fiber-based diet promotes higher levels of dietary microbial diversity since fiber fermentation produces more by-products compared to starch fermentation. This is consistent with the results of this study where forage nutrient quality decreased, while microbial diversity increased in WGY compared to GY and REGY. Compared to other factors such as the environment, rumen microbial diversity is strongly affected by diet. The results of this study show that during different forage growth stage, bacterial composition remains basically identical, however, the relative abundance of microbes differs. *Dan et al. (2016)* reported that the yak grazing on the QTP, *Bacteroidetes* and *Firmicutes* were the two predominant bacteria, representing 59.06% and 25.81% of all sequence, respectively. The same conclusion was obtained in the present study where the relative abundances of these two phyla reached 51.84% and 34.16%. The similar proportion in relative abundance of these two phyla may be attributed to same in the available nutrients (both in QTP). Nevertheless, previous studies demonstrated that the relative abundance of *Firmicutes* were higher than *Bacteroidetes* in yak and bison at grazing patterns (*Huang, Li & Luo, 2017*; *Noel et al., 2017*). The reason related to the low *Firmicutes* abundance in this study might be this phylum could be impacted by numerous factors, such as age (*Xufeng et al., 2015*), location (*Huang, Li & Luo, 2017*) and diet (*Pitta et al., 2010*). *Guo et al. (2008)* reported that a low-calorie diet that is rich in protein and fiber seems to favor *Bacteroidetes*. It is reported that artificial high-energy diets that are rich in starch and fat seem to favor *Firmicutes*. The same conclusions were achieved in the current study, where the content of CP was positively correlated with *Firmicutes* and had the lowest value in the WGY group. The relative abundance of *Firmicutes* during GY is higher than at other stages, which might be due to alpine meadow nutrients and yield reached peaks and animals quickly gained weight from July to October. *Xue, Zhao & Zhang (2005)* and *Ley et al. (2006)* showed that *Firmicutes* and body weight follow a specific positive correlation in the human gut microbiota, which might be part of the reason why *Firmicutes* had higher relative abundance in GY and lower relative abundance in WGY. *Huang, Li & Luo (2017)* reported the proportions of *Firmicutes* in sheep of the QTP were higher compared to those of low-elevation sheep and goats and that Gram-positive bacteria may play an important role in the digestion of specific grass on the QTP. This also explained the significant changes of *Firmicutes* in the three stages in the QTP. As in other studies of yak grazed on QTP, the three most abundant microbial phyla were *Firmicutes*, *Bacteroidetes*, and *Proteobacteria* (*Dan et al., 2016*; *Huang, Li & Luo, 2017*; *Xue et al., 2017*). However, in the current study, third most abundant phylum was *verrucomictobia* (5.97%), followed by *Proteobacteria* (1.92%). Studies have shown that *verrucomictobia* contains a broad repertoire of glycoside hydrolyze enzymes and make a significant contribution to polysaccharide and cellobiose degradation (*Zoetendal et al., 2003*; *Godoy-Vitorino et al., 2012*; *Gharechahi et al., 2015*). In this study, the higher relative abundance of *verrucomictobia* compared to other study might be caused by different grassland types. *Synergistetes* have been detected in mammalian intestinal tracts (*Shah et al., 2009*). However, the function of cellulose and xylan-pectin enrichments from *Synergistetes* was not clear enough. It might be the one reason that *Synergistetes* in the rumen had the low relative abundance in the current study. The phylum *Fibrobacteres* was considered to cellulolytic bacteria and shown to be influenced by fiber content in the diet

(*Fernando et al., 2010*; *Jami & Mizrahi, 2012*; *Paz et al., 2018*). Several studies of the rumen microbiome have suggested that the phylum *Fibrobacteres* varies considerably across cows and diets. It is reasonable that these phylum of different abundance in this paper due to different fiber content in the forage in three seasons.

At the family level, few differences were reported in the ruminal microbiota of dairy cattle fed with either grass or total mixed rations (*De Menezes et al., 2011*). The greatest differences were found in *Lachnospiraceae* and *Veillonellaceae* in this study. *Lachnospiraceae* found in Holstein cows rumen and play important role in fibrolytic bacteria (*Godoy-Vitorino et al., 2012*; *Ziemer, 2014*). The same conclusion was also reached in the current study, where the relative abundances of *Lachnospiraceae* reached 12.62% (REGY). *De Menezes et al. (2011)* also reported that *Veillonellaceae* were observed at levels three times higher in cows that were fed with forage grass, which corroborates the conclusion reached in this study, since these microbial families were found in large proportions in yaks that grazed on pasture. *Wang et al. (2019)* reported that many microbes that belong to the family of *Ruminococcaceae* are fibrous-degrading bacteria with the ability to degrade protein. In the current study, the lowest relative abundance of *Ruminococcaceae* was presented in WGY, which is consistent with the change of CP in three periods. Studies demonstrated that members of the *Bacteroidetes* phylum that contain a varied combination of CAZymes (Carbohydrate-Active Enzymes) in the form of PULs (polysaccharide utilization loci), which encode multiple proteins include the detection, sequestration, hydrolysis and transport of complex carbohydrates (*Gharechahi & Salekdeh, 2018*). It is reported that particular PUL can be used to predict the substrate specificity of microbial strains and the presence of many PULs in *Bacteroidetes* genomes indicates their broad substrate specifcity and high potential carbohydrate degradation ability (*Stewart et al., 2018*). Therefore, *Bacteroidales_BS11_gut_group*, *Bacteroidales_S24-7_group*, and *Bacteroidales_RF16_group* all account for a large proportion in this study due to their wide functions. The forage growth stage ($p = 0.015$, Fig. 5) and the gender of the animal (GY, $p = 0.018$, Fig. 5) all had significant effects on *Bacteroidales_S24-7_group* in the current study and their relative abundances were highest during GY (Table S3). Research by *Long et al. (1999)* showed that soluble sugars are susceptible to seasonal and regional effects and reached a maximum in July in the QTP region. *Stewart et al. (2018)* also identify RUGs with large numbers of PUL related to *Prevotellaceae* family that contain proteins capable of binding and digesting multiple carbohydrate substrates. Consistent with this, the large relative abundance of *Prevotellaceae* in the rumen of yak in this study indicated that these family related to the high potential carbohydrate degradation ability. Two recent studies reported that *Christensenellaceae*, *Ruminococcaceae*, *Rikenellaceae*, and *Prevotellaceae* may play important roles in forage degradation in the rumen since these groups tightly adhere to forage grass after incubation in the rumen (*Liu et al., 2016*; *Shen et al., 2017*). In addition, it has been reported that the abundance of *Christensenellaceae* is associated with changes in rumen pH (*De et al., 2016*).

At the genus level, bacteria of *Treponema* in the rumen are able to degrade plant polysaccharides from hay or from a concentrated diet. As one of the core members of the rumen bacterial community (*Bekele, Koike & Kobayashi, 2011*), the same conclusion

could also be reached in the current study, where the *Treponema_2* showed similar relative abundance during all three forage growth stages, with an average abundance of 0.7%. The genus *Butyrivibrio* plays an important role in the decomposition of urea, protein, hemicellulose, cellulose, and complex carbohydrates (*Peng et al., 2015*). In this study, the relative abundance of *Butyrivibrio_2* in the GY was significantly higher ($P < 0.05$) than in the other two periods. *Prevotella* is capable to metabolize dietary fibers from plant cell walls, and thus produces significant amounts of short chain fatty acids (SCFAs) that are later absorbed by the animals (*Ramayo-Caldas et al., 2016*). Furthermore, *Prevotella*, which participates in the processing of complex dietary polysaccharides (*Ellekilde et al., 2014*), may promote an increased uptake of monosaccharides in the host. The average abundance of *Prevotella* reached 15.62% in the current study. *Prevotella* participates in the degradation of cellulose to produce VFA and break down polysaccharides in the plant into monosaccharides, thus supplying the yak with energy. This results in its higher relative abundance during WGY due to the increased fiber and lignin contents as well as the low nutrient content in the alpine meadow. Consequently, yaks must increase their feed intake to fight cold weather during winter. *Prevotella* is a large genus with high species diversity and plays an important role in the degradation of cellulose and vegetable protein (*Ley et al., 2006*). *Clostridium* is responsible for the degradation of cellulose (*Burrell et al., 2004*) and is commonly found when studying yaks. However, in this study, the relative abundance of *Clostridium* was <0.05%, which may be due to the degradation of cellulose in yaks by other and currently undiscovered fiber-degrading bacteria or by the nutrients related to the forage grass (*Bergmann et al., 2015*). The current study reached the same conclusion as the previous publication by *Shen et al. (2017)*, which showed that the abundance of fibrolytic bacterial genera (e.g., *Ruminococcus* and *Butyrivibrio*) and potentially fibrolytic bacterial taxa (e.g., *Christensenellaceae-R-7*, unclassified *Ruminococcaceae*, and unclassified *Rikenellaceae*) all occupy a majority at the genus level.

*Bolnick et al. (2014)* have shown that when male and female rats are fed the same high-fat diet, they show changes in relative abundance of gut bacteria only at the species level. When there is a difference in diet, males and females show apparent differences. In the current study, no difference was applied in the diets between male and female yaks during the same period. Consequently, under the same dietary conditions, male and female yaks only showed differences in relative abundance during REGY, GY, and WGY (Fig. 5). *Abreu (2010)* reported that Toll-like receptors (TLR) as a receptors which can mediating innate immune responses, and related microbiota. *Liu et al. (2017)* showed that newborn lambs with concentrates starter feeding increased the colonic mucosal mRNA expression of TLR and expression of TLR was associated with changes in the abundances of some specific bacteria like *unclassified S24-7*. *Trevisi et al. (2014)* indicated that bovine rumen can receive the immune cells. The *Bacteroidales_S24-7 _group* was higher in female yak than male yak during GY, the newborn animals intestinal flora were more closely associated with maternal, so the high *Bacteroidales_S24-7_group* in female yaks in this harsh environment may be related to this but it has not been confirmed. In male yaks, the relative abundance of phylum *Bacteroidetes* (WGY), family *Prevotellaceae* (WGY), and *Prevotella_1* (WGY) was significantly higher ($P < 0.05$) than in female yaks. In the current study, gender factors

make it difficult to interpret these results, but when the diet is compared to other factors, the gender of the animal exerts a relatively small effect on the composition of the microbiota (*Bergmann et al., 2015*; *Bergmann, 2017*).

Microorganisms influence immune function, nutrient absorption, and even enzyme metabolism (*Martin et al., 2010*). Tax4Fun is a software package that predicts the functional capabilities of microbial communities based on 16S rRNA datasets and Tax4Fun provides good functional profiles obtained from metagenomic shotgun sequencing approaches (*Asshauer et al., 2015*). Nucleotide excision, RNA degradation, Peptidoglycan biosynthesis, Starch and sucrose metabolism, and Amino sugar and nucleotide sugar metabolism were the most abundant functional genes in this study. KOs related to ABC transporters were reported as the the largest known protein family and are ubiquitously found in bacteria, archaea, and eukaryotes (*Zeng et al., 2017*). It might be the one reason that these KOs could found in high abundance in the rumen microbiota in this study. Besides, *Guo et al. (2018)* have shown that pathway related to ABC transporters was enriched on yaks and sheep grazing in the QTP. *Hamana et al. (2012)* demonstrated that ABC transport function as a barrier to protect the bovine not to be hurt from the intake of toxic substances. It is reasonable that these KOs of high abundance due to yak's ability to adapt to bad weather led to the increased metabolic function of the ABC transporters. Two-component systems are perceptually developed signal transduction systems (*Podgornaia & Laub, 2013*; *Zeng et al., 2017*). In the current study, KOs include ABC transporters, starch and Sucrose metabolism, and Amino sugar and nucleotide sugar metabolism, all of which had high relatively abundance during REGY, which might be related to the high CP content at the alpine meadow during REGY. The results of level 2 KOs (Table S5) suggest that the pathways related to Membrane transport had high abundance, which might be related to the high altitude and cold climate in the QTP, because the nutritional composition of alpine meadow grass differs, which in turn affects both the diversity and function of the bacteria in the yak rumen. According to the heatmap of 20 KOs in the level 3 KEGG pathway (Fig. S2), the functional KOs of rumen microbiota varied considerably in REGY, GY, and WGY.

## CONCLUSIONS

This study reports the effect of the forage on rumen microbiol diversity in yak of the QTP. Different forage growth stages changed the composition, diversity, and function of rumen microbiota in yaks that grazed naturally without feed supplementation in the alpine meadow of the QTP. Changes in temperature directly affect plant productivity, which in turn affects animal microbiome in the body. This effect is not related to the gender of the yaks. More studies about fungal amplicon sequencing (ITS) are needed to explore and the relationship between the microbial changes at different forage growth stages and gender could be better illustrated. Furthermore, research about archaeal composition in the rumen of yaks are required to promote the understanding of the reduction of CH4 emission and can also enhance the livestock production efficiency.

## ACKNOWLEDGEMENTS

The authors would like to acknowledge the warm-hearted help of the research group during the sample collection and experimental data analysis.

### Funding

This work was supported by the National Key Research and Development Program of China (2016YFC0501905), ''Strategic Priority Research Program'' of the Chinese Academy of Sciences, Grant No. XDA20050104, the Science and Technology Program of Qinghai, China (2019-SF-149,2019-SF-153) and the Qinghai Innovation Platform Construction Project (2017-ZJ-Y20). The funders had no role in study design, data collection and analysis, decision to publish, or preparation of the manuscript.

### Grant Disclosures

The following grant information was disclosed by the authors:
National Key Research and Development Program of China: 2016YFC0501905.
Chinese Academy of Sciences: XDA20050104.
Science and Technology Program of Qinghai, China: 2019-SF-149,2019-SF-153.
Qinghai Innovation Platform Construction Project: 2017-ZJ-Y20.

### Competing Interests

The authors declare there are no competing interests.

### Author Contributions

- Li Ma performed the experiments, analyzed the data.
- Shixiao Xu conceived and designed the experiments, performed the experiments, approved the final draft.
- Hongjin Liu performed the experiments, analyzed the data, authored or reviewed drafts of the paper.
- Tianwei Xu performed the experiments, authored or reviewed drafts of the paper.
- Linyong Hu and Na Zhao contributed reagents/materials/analysis tools.
- Xueping Han prepared figures and/or tables.
- Xiaoling Zhang performed the experiments, prepared figures and/or tables.

### Animal Ethics

The following information was supplied relating to ethical approvals (i.e., approving body and any reference numbers):

Northwest institute of Plateau Biology, CAS-Institutional Animal Care and Use Committee provided full approval for this research (NWIPB20160302).

### Field Study Permissions

The following information was supplied relating to field study approvals (i.e., approving body and any reference numbers):

This project is carried out with the permission of the local government (Qinghai province, China), so official documents and field permits were not required in this study, the China Qinghai provincial science & technology department gave these authorizations to allowed us to carry out the experiment here.

## Data Availability

The sequencing data for the 16S rRNA genes are available in the NCBI Short Read Archive: PRJNA504932.

## Supplemental Information

Supplemental information for this article can be found online at http://dx.doi.org/10.7717/peerj.7645#supplemental-information.

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
