# Peer review of "Yak rumen microbial diversity at different forage growth stages of an alpine meadow on the Qinghai-Tibet Plateau"

_PeerJ, doi:10.7717/peerj.7645_

## Round 0.1 · original submission · Major Revisions

Please consider all reviewer comments before resubmitting. In addition, it is my feeling that your abstract is too long and should definitely stick to highlighting the major findings. As for the intro, this needs to focus on more of a "why do I care" aspect.The authors mention that yaks provide resources to local heardsman and that climate change affects their forage, but there should also be more about how these topics relate to overall research on ruminant microbial diversity. I did appreciate that this study was conducted over a full season, and only on grazing yaks. While this seems to fill a hole in current research, I still question the relevancy. Are most yaks in the Qinghai-Tibet Plateau not supplemented in their feed? Are there full season studies conducted on feed supplemented yaks? A little more background information could really help here. I was also confused as to why the you mentioned pyrosequencing in the intro, but then later say that they used an Illumina MiSeq for this study. It seemed like unnecessary information and makes for confusion.

Only ten yaks were used in this study, five of each sex. Is this enough to truly support statistical comparisons of the microbiomes between the sexes? Is that really enough to do PCA?

Tax4fun seems like an interesting and potentially useful function prediction tool. I was a bit confused on the Level 1, Level 2, etc. designations though. Knowing that there are “…35 Level 2 functional groups and 284 Level 3 functional groups…” doesn’t really help me if I don’t know what makes a function belong to a certain Level. A couple sentences on how the tool works might be useful here for clarification. The discussion section on the findings from Tax4fun also seemed to be lacking, given that the authors specifically mention it in the title. The end the section by saying, “However, the function and role of some microorganisms in the rumen of yak during different seasons are still unclear, and the changes in rumen microbes in yak should be further studied.” so I was left wondering how useful this tool actually was for the paper’s objectives.

Tables 2-5 would have been much better as supplementary data. They take up a huge amount of space in the manuscript and don’t do a good job highlighting general trends. Figures (perhaps a stacked bar graph or bubble plot) could convey the information in a more concise way. Figure and table captions were generally lacking. Figures should be stand-alone. The reader shouldn’t have to look through the body of the text to find basic information (e.g., abbreviations of grass stage names). Text on the figures was also too small. Formatting in the figure captions was inconsistent with the text (e.g., “STAMP” was lowercase in Figure 3 caption, but uppercase in body of text).

While I wish to be sensitive to the idea that English may not have been a first language for the authors, the entire paper contained awkward language structure (e.g., paragraphs are stand-alone and don’t flow together well) and formatting issues (e.g., multiple instances of no spaces before and after commas). Because of this, I felt that a lot of the major points and conclusions were lost among the supporting information. Although the science itself seems to fill a gap in the current research, the writing weakens the arguments and the evidence presented.

·

Basic reporting

The authors do an excellent job of introducing their model system and ruminant and frame the relevance of their study well. Readability is reasonable, but the text contains a small, but significant, number of typos and formatting issues including spacing near commas. I would recommend these be addressed prior to acceptance. Figure quality is sufficient for publication. Availability of data fits acceptable standards, but fastq files would be preferable.

Experimental design

The manuscript fits within the journal scope and addresses an area of limited knowledge; yak ruminal ecology across seasons on pasture with distinction between sexes. The experimental design is limited to the harsh environmental constraints of the system studied and contains replicable detail. From the text it appears 10 experimental animals (even sex-split) were grazed for nine months in a herd of 30 animals. These parameters are sufficient to support the authors’ conclusions regarding rumen microbial ecology and function. Sample collection and storage protocols followed well-established norms.

Validity of the findings

Presented data and statistical treatment of data are robust. The authors address both the amplicon-derived ecology and taxonomy-predicted function of the yak rumen microbiome as well as forage nutritional composition in great detail. As a largely discovery-oriented, incremental manuscript, the authors provide a baseline reference for future inquiries into yak microbial ecology or the effects of range seasonal phenology on the yak rumen microbiome.

Additional comments

I believe future work would benefit from the inclusion of fungal amplicon sequencing (ITS).

Your treatment of the microbial sequence data is robust, but I would recommend exploring Love et al., 2014 for the purposes of differential abundance between groups and Koh et al,. 2017 for relating microbial data to continuous covariates, like forage quality indicators, if you have not already done so.

Reviewer 2 ·

Basic reporting

We know that It is difficult to do experiment use yaks in Qinghai Tibetan plateau. However the importance of the present experiment is not so high, because we know that when the diet changes the microbiota in the rumen change accordingly. We do research we can come to a conclusion that will guide the herdsman to farm yaks and increase economic income or protect the grassland, however neither the aims come true.

Experimental design

It is said in line 87 that ten healthy yaks were selected, then in line 88 thirty cattle without supplementary feeding were allowed to graze, I do not know how the experiment was designed.

Validity of the findings

The importance of the findings of the study is not so high I think.

Additional comments

Line 13 and line 31 P<0.05 P<0.05 P < 0.05 , please keep the format in line with the requirement of the journal. And other places in the whole paper. Line 285 reference et al or et al ? check the whole paper please.
Line 50 line 54 reference & or and ? check the paper carefully other places
Line 100 24 hours or h ?
Line 105 you did not show the pH value of the rumen liquid, it is not necessary to declaim how to determine the pH value.
Line 111-112 how to determine the CP EE ADF NDF, add the references ! AOAC ? id 984.02 ? van soest 1991 to determine NDF ADF
Line 133-134 I suggest you use SAS to analyze data and do not use GLM model.
Table 1 you should present SEM and P value of data
Tabel 2-7 why you present reg (%) g (%) in the tables ?
Table 3-5 the phylum family and genus, the importance of the three tables are not so high, why you do not present order class ? or just delete family.
Fig 4 why you only present the difference between female and male ? what about the difference among three seasons you investigated, that is the important point ?

Annotated reviews are not available for download in order to protect the identity of reviewers who chose to remain anonymous.

Reviewer 3 ·

Basic reporting

The study is both interesting and focused on the real changes of the rumen in the animals studied. However, the structure and presentation of the manuscript could be better if the authors improve several items.
Tittle should emphasize characteristics of the study more than the techniques because it is not a paper describing a new methodology, it is a paper that describes microorganisms. It would be better to consider the species of the animal in the title, but in the paragraphs use the word yak. The phrase ‘during different phenological periods in the montane grasslands of the Qinghai-Tibet Plateau’ of the line 10-11, it is adequate to be included in the title.
Authors did not select the best keywords for the study. For example, the word forage had to be considered. I suggest making a new selection of keywords.
Regarding introduction, the authors did not describe in detail previous studies about the abundance of microorganisms in the rumen of yaks. Perhaps the information of the rumen of the animal living in the same zone could support the reason of the study.
Regarding grammar, authors used capital letters after a comma, did not respect the space after a full-stop, the expression et al was not written in cursive. Sometimes they used verbs in present when they should be written in past. Authors should revise and correct the entire manuscript.
Line 18 The most abundant phyla were, not are.
Line 18- 19 The phyla Bacteroidetes and Firmicutes are mentioned twice.
Line 20 The word respectively is not adequate in this context. Authors should mention all the percentages related to the phyla or re-write the sentence.
Line 26-27 The ruminal microbial flora was, not is.
Line 30 The word abundance would be better in this case, you use the word changes twice in the same statement.
Line 42 It is not clear if this climate change is referred about differences in the weather, or in the process that is still in discussion if it is real o not.
Line 46 Components of plants affect the nutritional state, and this situation affects animal growth.
Line 49 Archaea do not produce volatile fatty acids, they use them to produce methane. Authors should consider the real role of the microorganisms in the rumen.
Line 53-64 Authors should mention the microorganisms that have been detected in previous studies.
Line 62 If there are not several studies of the microorganisms in rumen, authors should emphasize the use of high- throughput sequencing of their research. DNA pyrosequencing is related to the platform 454 of Roche. Currently, there are several platforms to sequence DNA of microbial communities. The word high- throughput sequencing is better to use in this case.
Line 69 Define what is a short period of time would help to support the importance of to do a study for a long time.
Line 70 Provide a short definition of phenological periods.
Line 71 Delete the word therefore.
Why did the authors select only forage as food in the study? It is not clear if the animals can eat other different kinds of food.

Experimental design

Authors mentioned the three stages of the forage: regreening, grass and withered grass. It would be better the use of abbreviations for these words in all the manuscript. In addition, the description of the information they would be to use the same order to describe the three stages. For example, first they could mention to regreening, then grass and the last withered. These suggestions to make easier to read the manuscript since in some paragraphs is a little difficult. If the authors possess photographs of the forage in the stages mentioned that could support the idea mentioned above.
Line 88 Which was the role of thirty cattle?
Line 94 To know the proportion (in percentage) of every species could be interesting for the discussion of the results. It will be adequate to consider if it has been described the components of the walls of the plants that constitute the forage and to find a relation with the results obtained in the prediction and the abundance of the microorganisms.
Line 96- 101 Authors mentioned the sessions in the abstract, they should relate samples with them in this part of the methodology.
Line 105 The stomach of ruminants possesses different zones. It would be interesting to know the zones that have been described in yaks and where samples were taken. It is not clear if the methodology to take samples was described previously or it was a procedure implemented in this study.
Line 106 Begin the statement with the word twenty, not 20.
Line 109 How many measurements did the authors do?
Line 115 Mention the number of samples that were analysed.
Line 117 If you analyse bacteria, you analyse prokaryotic no eukaryotic.
Line 119 It is necessary to mention the reference of the primers.
Line 124 It is necessary to mention the database used to do the taxonomic classification.
Line 132 Add the title; statistical analyses.

Validity of the findings

Authors presented Venn diagram, but they do not describe in the results the figure.
The tables present the term pooled-SEM or observed species that were not mentioned in the manuscript.
Authors did not present bar-charts of the relative abundance that would help to understand better the results.
Some statistical analysis can compare physicochemical parameters and abundance of microorganisms, such as Canonical Correspondence Analysis, that were not considered and could give information of the most important components of the forage that had influence in the microorganisms.
The prediction of the metabolic function is adequate; however, I consider the focus in the abundance of microorganisms is most important because they are a real result, not a prediction.
Line 138- 148 Authors mentioned the significant differences of the results; however, they did not compare the differences among the samples. For example; a presented 10% more protein than B. Authors should rewrite the paragraph and consider the differences detected among the samples.
Line 146- 148 If authors know the percentage of each plant, they should incorporate the information into tables.
Line 152 To describe in methodology what animals were selected. In methodology is only specified 10 yaks and another group that was not described in detail.
Line 160 It is better to use the word gender in this case, not sex.
Line 160-162 If the gender of animal presented differences, this would be part of the abstract.
Line 171 Why did the authors consider only the 15 most abundant species? Microbial communities always possess microorganisms in a low abundance as well as uncultivated microorganisms. It is adequate to mention them because they can possess a high activity and an important role in the degradation of the organic matter, even though they are not present in a high abundance or their function is not known.
Line 185 It is better to write; of different genera, no sexes.
To relate genera with phyla could help to understand the taxonomic groups that present more changes. Families would also be associated to a phylum.
Line 253- 263 This paragraph would be in the introduction because it is the reason of the selection of specific times to take samples.
Line 274 There are previous studies that describe pathways to degrade cellulose in Firmicutes and Bacteroidetes.
Line 294 Synergistetes is related to syntrophic associations with archaea to produce methane.in anaerobic environmental.
Line 312 What is the meaning is sex x stage?

Additional comments

no comment

---

## Round 0.2 · Major Revisions

Please attend to the many efficacious suggestions made by the reviewers. You should also document your disposition on each of these suggestions, if you chose to resubmit.

·

Basic reporting

The authors use clear, professional English to convey a thorough background literature introduction, methods used, results obtained and a discussion of findings. Article formatting, figures, tables, and raw data are sufficient for publication. In the future, please always provide sequence files as ".fastq" (fasta files with base quality information) rather than ".fasta" (no base quality information).

The conclusions of the study are well-aligned with the scope of the work. Namely, that this is an initial exploration (small N) of seasonal and sex-based changes in yak rumen microbiomes on QTP range and that findings are of primary utility to the development of more robust future inquiry.

Experimental design

The presented work is original and within scope.

The research question presented is "how do bacterial composition and bacterial composition-inferred function of the yak rumen differ across season and sex on the QTP?" This question is presented as meaningful given the importance of yak as livestock to the agricultural region of the QTP and the importance of the rumen microbiome to efficient production as detailed in non-yak ruminants.

The methods used meet technical and ethical standards.

Technical methods are described in sufficient details as to be replicable.

Validity of the findings

Given the largely preliminary nature of the study and the complexities of long-term rangeland-based animal trials, I believe that the design is sufficiently robust to support the authors' conclusions.

Figures and data included in the submission appear to match and are treated in a statistically sound manner with the notable exception of the NCBI accession quoted under the data availability header. SRP 169600 is not associated with your study as far as I can tell. This must be amended.

Conclusions made are not overstated, address the research question, and identify future directions based on challenges discovered in the course of study.

Additional comments

I acknowledge the challenges the authors faced in conducting a long-term on-range trial. I hope to see the lessons learned from this initial investigation applied to focused future work with QTP yak livestock.

Reviewer 3 ·

Basic reporting

The manuscript presents an improvement in relation to the first version. However, there are some parts that it is necessary to change. In general; methodology, results and discussion were improved; however, the introduction has some grammar mistakes and it is essential to develop in detail the importance of the study and its new contribution. There are some names of phyla and expression ‘et al’ that are not written in cursive, authors must change them.
The abstract should be re-organized. It is necessary to write the importance of the study before the overall
Line 23 In this study, the diversity… this sentence is methodology should be after the overall.
Line 26-28 This work investigated the rumen bacteria… this sentence is the overall and should be introduced in line 23, after the sentence that finishes in 3000- 5000.
Line 33 The word strongly is not necessary for the description.
Line 34 Bacteroidetes and Firmicutes must be in cursive script.
Line 36-37 Authors did not try to separate samples. It is better to write that taxonomic groups did not present differences regarding gender.
Line 38-39 Re-organize the sentence like this; The rumen microbiota was enriched with functional potentials that were related to ABC transporters, the Two-component system, Aminoacyl-tRNA biosynthesis, and metabolism of Purine, Pyrimidine, Starch and sucrose metabolism.

Line 46 QTP is enough, this abbreviation was mentioned in the abstract.

Line 48 the harsh environment of the QTP, with its low oxygen… the harsh environment of the QTP that possesses low oxygen…

Line 49 Natural alpine meadows in the QTP have qualities… Natural alpine meadows in the QTP also have qualities

Line 51 Change: The growth period of grass is approximately 100 to 150 d per year, to The growth period of grass is approximately from 100 to 150 d per year.

Line 70 Previous studies have shown…

Line 71 and seasons all impact on ruminal microbes…

Line 73-74 (Xue et al.,2017) reported… in this sentence, authors need to write only the year in parentheses. Xue et al. (2017) reported.

Line 81 In this paragraph, the authors could mention some taxonomic groups detected in yak rumen.

Line 88 High-throughput sequencing technology has provided microbial…

Line 89 Change the word and for as well as.
Line 90-92 This is a short description of the research, but it is necessary to write the importance of this study and its contribution.

Experimental design

Authors improved considerably the methodology, however, there are some mistakes that it is necessary to change.

Line 96 besides this project is… besides this project was
Line 99 Change word granted to granting.
Line 102 sample plot is 3265 m… sample plot was 3265 m.
The sample plot has… The sample plot had
Line 103 Delete comma
Line 103-106 Was the place´s information monitored by the authors? or was information provided by a special office? If it was provided for other office or research group, authors should mention it.
Line 108- 109 These ten yaks without supplementary feeding during the experimental stage were allowed access to water … These ten yaks, without supplementary feeding during the experimental stage, were allowed access to water.
Line 113-119 If authors monitored these percentages, these values they would be in results as a supplementary table. In the case that authors receive the information of another group, they should mention it after the description of the forage.
Line 125 to investigated… to investigate.
Line 130-132 Change: This method has been used in previous studies (Sun et al., 2018; Xue et al., 2017) and (Ramos-Morales et al., 2014) have shown that stomach tube and rumen cannulation methods do not result in changes of rumen microbiota, to ´This method has been used in previous studies (Ramos-Morales et al., 2014; Xue et al., 2017; Sun et al., 2018), and they have shown that the stomach tube and rumen cannulation methods do not have any influence on the results´.
References must be organized from the older to the younger.
Line 136 Delete word ´were´.
Line 143 If authors used stool samples, it means that they studied gut microbiome. The manuscript is focused on rumen microorganisms, authors should delete or change this word.
Line 151 If you use a sequencer that was in your laboratory, this statement is right. If you take the service of a company, you should mention the place where the samples were sequenced.
Line 154 Version of software is enough, it is not necessary to write the web page.
Line 163- 164 In this case, reference is adequate, it is not necessary to write the web page.
Line 171 Mentioning the web page is right in this case.
Line 177 web page was mentioned before, it is not necessary to mention it again.

Validity of the findings

Description of the results was improved, there are some mistakes that it is necessary to change.


Line 192 The CP content was highest… The CP content was the highest
Line 199 Authors should write a short conclusion of this table.
Line 205 Authors described 65,92% of total OTUs, there are three zones that were not mentioned; 1599, 1297 and 1038 correspond to 34.08% of OTUs. Authors should add information about them.
Line 210 with highest number… with the highest number
Line 211- 212 with highest number… with the highest number
Line 214- 217 I suppose these values are the average of the three values detected in samples, authors should mention it in the paragraph.
Line 222 Firmicutes in cursive script.
Line 224 Proteobacteria in cursive script.
Line 228 Firmicutes in cursive script.
Line 269 Change word furthermore for another, such as in addition or moreover, authors used this word at the beginning of the paragraph.
Line 278 The sentence: female could not be separated from male samples should be improved, authors described a process, it is better to write that it was not detected differences in PcoA between female and male samples.
Line 294 The word but is informal, it is better to use the word, however.
Line 319 Which is the difference between low and high-quality diet? Is it about different kind of nutrients, food or plants?
Line 325- 328 This information could be in the introduction. In this part, it would be appropriated to write results obtained in this study are according to previous studies.
Line 330 Authors should emphasise the importance of obtaining similar abundance in different studies since this indicates the relevance of both phyla in the degradation of the plant cell wall.
The study mentioned, was it about yak or another animal?
Line 331 Bacteroidetes in cursive script.
Line 332 Firmicutes in cursive script.
Line 334 The relative abundance of Firmicutes in the WGY group was lower
335 than in REGY and GY; this sentence describes a result. It should not be considered in this part of the manuscript.
Line 340 You should write: Xue et al. (2005) and Ley et al. (2006) showed...
Line 342 Delete the word ´but´, the word ´and´ is better in this case.
Line 348- 350 If authors detected phylum Synergistetes, this sentence should be related to the results of the study. However, if the authors did not detect this phylum, they should discuss why they did not detect it. In addition, anaerobic environments possess several phyla that authors could consider in the discussion since they could be in these animals.
Linea 353 Where were these families found? Yaks? Humans?
Line 356 change to: de Menezes et al. (2011).
Line 360 Delete the dot after the word In.
Line 362 The pathways to degrade plant cell wall have been broadly studied in Bacteroidetes during the last years. Authors could improve the discussion using this information.
Line 369 Research by (Long et al., 1999) … Research by Long et al. (1999)
Line 401 (Bolnick et al. 2014) have shown… Bolnick et al. (2014) have shown
Line 402 Rats are not a ruminant animal. Perhaps the authors wanted to describe other microorganisms in rat’s bodies.
Line 423-430 This sentence is very long and difficult to understand. Authors should write their ideas newly.
Line 442 This study reports the effects of on rumen microbial diversity… this study reports the effect of the forage on rumen microbial diversity
Line 446 that the gender of the animal had less impact on yak ruminal microorganisms… this effect is not related to the gender of the yaks.
Line 446-449 Authors should also consider the study of archaea in conclusions.

Additional comments

No comments

---

## Round 0.3 · Minor Revisions

The SRA accession number still needs to be added to your manuscript in the materials and methods section. Also, line 165, should read "method". Finally, in line 270 and line 288, please remove the word "more". Please make these corrections and resubmit.

---

## Round 0.4 · accepted · Accept

Thank you for getting the accession info added. I see that it is now available through NCBI's short read archive.